# The Vascular Endothelial Growth Factor-A121/Vascular Endothelial Growth Factor-A165 Ratio as a Predictor of the Therapeutic Response to Immune Checkpoint Inhibitors in Gastric Cancer

**DOI:** 10.3390/cancers16233958

**Published:** 2024-11-26

**Authors:** Yuki Hamada, Kiyonori Tanoue, Takaaki Arigami, Munekazu Yamakuchi, Masashi Okawa, Daisuke Matsushita, Kazunori Takenouchi, Shingo Yamada, Drew N. Maywar, Chieri Nakayama, Yoko Oyama, Sadayuki Higashi, Chieko Fujisaki, Yuto Hozaka, Yoshiaki Kita, Teruto Hashiguchi, Takao Ohtsuka

**Affiliations:** 1Department of Digestive Surgery, Graduate School of Medical and Dental Sciences, Kagoshima University, Kagoshima 890-8544, Japanarigami@m.kufm.kagoshima-u.ac.jp (T.A.);; 2Department of Laboratory and Vascular Medicine, Graduate School of Medical and Dental Sciences, Kagoshima University, Kagoshima 890-8544, Japan; 3Department of Cardiovascular and Gastroenterological Surgery, Graduate School of Medical and Dental Sciences, Kagoshima University, Kagoshima 890-8544, Japan; 4Shino-Test Corporation, Sagamihara 252-0331, Japan; 5Department of Electrical and Computer Engineering Technology, Rochester Institute of Technology, Rochester, NY 14623, USA

**Keywords:** vascular endothelial growth factor-A121, VEGF-A165, gastric cancer, immune checkpoint inhibitor therapy

## Abstract

As vascular endothelial growth factors (VEGFs) are closely associated with intra-tumor immunity, we examined whether the value of VEGF isoforms could predict the efficacy of immune checkpoint inhibitors (ICIs). The ratio of serum VEGF-A121 to VEGF-A165 significantly prolonged recurrence-free survival in the low value group and was also associated with treatment response. This study indicates that the serum VEGF-A121/165 ratio may be an easily measured new biomarker for predicting treatment responses to ICIs.

## 1. Introduction

The use of immune checkpoint inhibitors (ICIs), including inhibitors of the programmed cell death protein pathway (PD-1/PD-L1) and cytotoxic T-lymphocyte antigen 4 (CTLA-4), have greatly influenced the treatment of many cancers. Although the response rates to these drugs in monotherapy are relatively low, with a range of 15–20%, few patients achieve a durable complete response [1,2,3,4]. Identifying biomarkers for predicting the response to ICI treatment is currently the most critical aspect of this therapeutic strategy. PD-L1 expression, which is measured via immunohistochemistry, serves as a biomarker currently used in the clinical setting, and it may be useful in selecting candidates who are potentially responsive to ICI treatment. However, the correlation between the detection of PD-L1 expression in tumors and overall response rate (ORR) is inadequate [5], and its utility as a marker of PD-L1 expression has been debated [6,7]. Tumor mutation burden (TMB), which is a measure of the number of mutations in cancer, and microsatellite instability (MSI), which is associated with DNA mismatch repair deficiencies, are also expected to represent predictive markers of ICI response. Several studies have reported an association of high TMB and MSI with ICI responses [8,9,10]. However, these markers have not been converted into definitive and clinically suitable biomarkers [11,12].

Recently, ICI treatments have been used in combination with other drugs, chemotherapies, and molecular-targeted therapies. Of these drugs, vascular endothelial growth factor (VEGF)/VEGF-receptor inhibitors exert the most promising effects on the immune response in combination with ICI treatment, with additive and synergistic effects. VEGF is associated with tumor angiogenesis as well as the tumor immune microenvironment. Several clinical trials have reported the favorable antitumor activity of anti-PD-1 antibodies in combination with VEGF/VEGF-receptor inhibitors [13,14]. In particular, the combination of ICIs and VEGF inhibitors shows good results against advanced hepatocellular carcinoma (HCC) [15], and the combination of atezolizumab (anti-PD-L1 antibody) plus bevacizumab (VEGF inhibitor) is now considered a first-line treatment for patients with advanced HCC. The role of VEGF in the immune microenvironment of tumors has been the focus of considerable attention, and the potential of VEGF as a predictive marker of therapeutic response in immunotherapy has been explored [16,17].

VEGF-A, which is mainly involved in tumor angiogenesis, has several isoforms, each of which affects angiogenesis via a different function [18,19]. However, the utility of VEGF isoforms as an indicator of therapeutic response to ICIs has not been determined due, in part, to the lack of an accurate assay. Therefore, our objectives were to examine the isoforms of VEGF and determine whether VEGF levels predict ICI efficacy. Recently, we developed and reported a novel assay system for measuring levels of VEGF-A121 and VEGF-A165, the major isoforms of VEGF-A [20]. Here, we leverage this assay to measure levels of the isoforms of VEGF-A in the serum of patients with gastric cancer treated with ICIs and verified their association with the therapeutic effect.

## 2. Materials and Methods

### 2.1. Ethics

This study was performed according to the Declaration of Helsinki and was approved by the Ethics Committee of Kagoshima University Hospital (approval number 180213). All patients who participated in this study were given full and accurate explanations, and their written consent was obtained.

### 2.2. Sample Collection

The medical records of 60 patients with unresectable or recurrent advanced gastric cancer who received monotherapy using nivolumab (anti-PD-1 antibody) between October 2017 and July 2020 at the Department of Digestive Surgery, Kagoshima University Hospital, were retrospectively reviewed. Of these patients, we selected 30 who received at least two regimens of standard chemotherapy for gastric cancer and had an Eastern Cooperative Oncology Group performance status (PS) of 0 or 1, according to the patient selection criteria of the double-blind randomized phase III trial of nivolumab (ATTRACTION-2) [21]. Serum samples were collected before anti-PD-1 antibody treatment (within 24 h) and stored at −60 °C until further processing at the Department of Digestive Surgery, Kagoshima University Hospital.

### 2.3. Measurement of Target Protein Levels in Serum

The levels of target proteins in serum, including total-VEGF-A, VEGF-A121, VEGF-A165, VEGF-C, PD-1, and PD-L1, were measured and validated via an enzyme-linked immunosorbent assay (ELISA). The total-VEGF-A, VEGF-C, PD-1, and PD-L1 levels were measured using commercial ELISA kits (R&D Biosystems, Minneapolis, MN, USA), and the VEGF-A121 and VEGF-A165 levels were measured according to our previous study [20]. We had measured VEGF-A121 and VEGF-A165 serum levels in healthy controls in that study and set the values found in that study as a baseline.

### 2.4. Evaluation of Clinical Indicators and Statistical Analysis

The appropriate cutoff points for the values of all parameters in the serum were defined based on the median value for all patients. Medical records were reviewed for patient characteristics and other laboratory results. The prognosis was verified using the Kaplan–Meier method for overall survival (OS) and progression-free survival (PFS) after the initiation of nivolumab treatment. The tumor response to the treatment was objectively evaluated using the Response Evaluation Criteria in Solid Tumors (RECIST v1.1), and the parameters that were shown to be associated with prognosis were validated in association with the treatment response. In addition to serum VEGF, PD-1, and PD-L1 levels, the age, pretreatment PS, endoscopic findings, histology, history of ramucirumab (VEGF-receptor inhibitor) treatment, metastatic lesions (liver, distant lymph nodes, and peritoneum), and the presence of ascites were evaluated as covariates of clinical findings to validate their association with prognosis using a univariable Cox proportional hazards model. Univariable analysis of the association with treatment response was performed via logistic single regression analysis. Clinical laboratory data such as modified Glasgow Prognostic Score, neutrophil/lymphocyte ratio, platelet/lymphocyte ratio, and lymphocyte/monocyte ratio values were also included in these analyses. Multivariable analyses were not performed owing to the small sample size of the study population. Fisher’s exact test was used to compare response rates between groups. All *p*-values were based on two-tailed hypotheses, and values of *p* < 0.05 were considered statistically significant. All statistical analyses were performed using JMP Pro software, v14.0.0 (SAS Institute, Cary, NC, USA).

## 3. Results

### 3.1. Clinical Characteristics of Subjects

A total of 30 patients with PS 1 or less were evaluated in this study, and the clinical characteristics are summarized in Table 1. The median age was 65 years and the range was 35–78 years. Of the 30 patients, 22 (73.3%) were male, 11 (36.2%) had a PS of 0, 19 (63.3%) had a PS of 1, 5 (16.7%) had endoscopic diagnosis of type 4, 23 (76.7%) had undifferentiated histology, and 11 (36.7%), 6 (20%), and 11 (36.7%) had peritoneal dissemination, liver metastasis, and distant lymph node metastasis, respectively. The median number of courses of nivolumab treatment was 5.5 (1–26); a total of 7 (23.3%) patients had received three or more courses of chemotherapy before nivolumab treatment, and 27 (90%) patients were treated with ramucirumab. Non-informative censoring occurred for 11 patients in the analysis of OS and 5 patients in PFS. Our study placed the 30 patients into two groups based on their level of each parameter; the 15 patients with the highest level of each parameter were placed in the high-serum level group.

### 3.2. Statistically Significant PFS Difference for VEGF-A121/165 Ratio

The overall survival (OS) of the low- and high-serum VEGF groups was assessed in terms of various molecular markers using the standard Kaplan–Meier method and summarized in Figure 1. The median OS time, taken as the effect size of OS, was measured to be prolonged for the low-serum group relative to the high-serum group for VEGF-A total (+26 days to 478, Figure 1A), VEGF-A121 (+68 days to 454, Figure 1B), VEGF-A165 (+181 days to 631, Figure 1C), VEGF-A121/165 ratio (+68 days to 454, Figure 1D), PD-1 (+105 days to 471, Figure 1F), and PD-L1 (+3.54 days to 454, Figure 1G), and was measured to be shortened in VEGF-C (−76.5 days to 386, Figure 1E). However, the log-rank test between the two groups yielded a *p*-value > 0.05 for all molecular cases, with the lowest being VEGF-A total (*p* = 0.29), VEGF-A121 (*p* = 0.32), VEGF-A165 (*p* = 0.42), and VEGF PD-1 (*p* = 0.42), and the highest being VEGF-A121/165 ratio (*p* = 0.55), VEGF-C and PD-L1 (both *p* = 0.86).

The progression-free survival (PFS) of the low- and high-serum VEGF groups was also assessed in terms of various molecular markers using the standard Kaplan–Meier method; the results are summarized in Figure 2. The median PFS time was measured to be prolonged for the low-serum group relative to the high-serum group for the VEGF-A total (+53 days to 118, Figure 2A), VEGF-A121 (+53 days to 118, Figure 2B), VEGF-A165 (+37 days to 114, Figure 2C), and VEGF-A121/165 ratio (+67 days to 129, Figure 2D), and was measured to be shortened in VEGF-C (−40 days to 74, Figure 2E), PD-1 (−15 days to 94, Figure 2F), and PD-L1 (−15 days to 94, Figure 2G). The log-rank test between the two groups yielded a *p*-value < 0.05 for the VEGF-A121/165 ratio (*p* = 0.0088), making this marker the only one to exhibit a statistically significant difference between the PFS of the two groups. All other makers exhibited a *p*-value > 0.05, with the lowest being VEGF-A total (*p* = 0.15), VEGF-A121 and VEGF-C (both *p* = 0.19), and the highest being VEGF-A165 (*p* = 0.87), and VEGF PD-1 (*p* = 0.65), and PD-L1 (*p* = 0.71). Univariable analysis via Cox hazard analysis showed that the serum VEGF-A 121/165 ratio alone was significantly associated with a relatively shorter PFS (hazard ratio (HR) = 1.08, 95% confidence interval (CI) [1.00, 1.17], *p* = 0.0454; Table 2).

### 3.3. Statistically Significant Association Between VEGF-A121/165 Ratio and ICI Treatment

When comparing the treatment response in the low- (*n* = 15) and high- (*n* = 15) serum VEGF-A-121/165 ratio groups, the RECIST assessments were 3 vs. 0 for complete response (CR), 2 vs. 0 for partial response (PR), 3 vs. 2 for SD, and 3 vs. 10 for PD. Patients with clinically undecided PR or SD were classified as non-CR/non-PD (4 vs. 3). The disease control rate (DCR) was 80% vs. 33% (*p* = 0.0253; Table 3). Further assessment focused on patients with the target disease (low (*n* = 8) vs. high (*n* = 11)): CR was 2 vs. 0, PR was 2 vs. 0, SD was 3 vs. 2, PD was 1 vs. 9, ORR was 50% vs. 0% (*p* = 0.0181), and DCR was 88% vs. 18% (*p* = 0.0055; Table 3). Univariable analysis with logistic single regression analysis was used to verify the association between each parameter and treatment effect (Table 4). Serum total-VEGF-A (risk ratio (RR) = 1.00, 95% CI [1.00, 1.01], *p* = 0.0032), serum VEGF-A121 (RR = 1.00, 95% CI [1.00, 1.01], *p* = 0.0043), and serum VEGF-A121/165 ratio (RR = 1.2, 95% CI [1.01, 1.38], *p* = 0.0319) showed statistically significant differences.

## 4. Discussion

The VEGF plays a critical role in the cancer microenvironment by promoting tumor angiogenesis; however, it is also implicated in having immunosuppressive effects [22,23,24,25,26,27]. The VEGF attenuates effector T cell activity along with dendritic cell differentiation and maturation, and it enhances the activity of regulatory T cells (Tregs) and myeloid-derived suppressor cells (MDSCs). We previously demonstrated that the suppressive immune environment in tumors is improved after the addition of VEGF inhibitors using resected specimens from patients with colorectal cancer [28]. Thus, the impact of the VEGF on the immune environment has led to the investigation of the possibility that its serum level may serve as a predictive marker for the therapeutic response to ICIs. In a previous study, high-serum VEGF levels were correlated with poor clinical outcomes in patients with melanoma treated with anti-CTLA-4 antibodies, and patients with high-serum VEGF levels had a significantly lower rate of disease control [16]. In patients with advanced non-small cell lung cancer aged ≥75 years, the high-serum VEGF group has been reported to have a lower ORR and DCR than those of the low-serum VEGF group, indicating that serum VEGF level is a predictive biomarker for the response to anti-PD-1 antibody treatment [17].

At least five types of VEGFs have been identified: VEGF-A, -B, -C, -D, and placental growth factor [29,30]; VEGF-A and -B have isoforms with splicing differences [31]. In gastric cancer, VEGF-A, -C, and -D are mainly associated with pathogenesis [32]; their role as biomarkers has been investigated in clinical trials such as AVAGAST [33], RAINBOW [34,35], ATTRACTION-2 [21], and ATTRACTION-4 [36]. Although validated in these clinical trials, there was no correlation with the therapeutic response. Thus, there is no consensus on the use of VEGFs as predictive markers of ICI treatment responses. Among these VEGFs, VEGF-A, which is mainly implicated in angiogenesis in tumors, has been investigated the most in the search for biomarkers for cancer therapy. However, most existing assays measure the combined total amount of VEGF-A121 and VEGF-A165, which are the major isoforms [20]. In the present study, we separately measured the amounts of VEGF-A121 and VEGF-A165 in ICI-treated patients and found that the VEGF-A121/A165 ratio was significantly associated with progression-free survival (PFS). In addition, a comparison of the low and high VEGF-A121/165 groups showed that the treatment was more effective in the low VEGF-A121/165 group.

VEGF-A121 and VEGF-A165 are the main isoforms expressed in most human tissues and cancers [37]. VEGF-A121 expression is higher than that of VEGF-A165 in cancer tissues obtained from patients with colorectal cancer, and tumors with high VEGF-A121 expression show increased tumor vascular distribution [38]. Plasma VEGF-A121 levels in patients with recurrent glioblastoma are higher than those in healthy controls, and VEGF-A121 levels are associated with prognosis in these patients [39]. In contrast, VEGF-A165 is correlated with prognostic deterioration in ovarian cancer [40]; both VEGF-A121 and VEGF-A165 are implicated in cancer development and progression.

VEGF-A121 and VEGF-A165 contribute differentially to cancer angiogenesis. Yu et al. revealed remarkable isoform-specific differences in vascular pattern formation in malignant tumors in a human melanoma model: VEGF-A121-overexpressing tumors are poorly angiogenic and necrotic, whereas VEGF-A165-expressing tumors have a high density and show an extremely heterogeneous formation of vascular networks. This indicates that VEGF-A165 expression provides the most effective tumor perfusion to cancers [19].

Neovascular vessels in tumor tissues have various structural abnormalities that inhibit immune cell infiltration compared to that in normal vessels; however, in tumor tissues, immune cells infiltrate into the tumor through the neovasculature [41]. Thus, aggressive antiangiogenic therapy promotes hypoxia by pruning tumor vessels and induces the formation of an immunosuppressive tumor microenvironment that suppresses the therapeutic effect of ICIs by recruiting Tregs, MDSCs, and M2-type tumor-associated macrophages [42]. The serum VEGF-A121/165 ratio may reflect the intra-tumor blood circulation status owing to differences in angiogenesis. Patients with a high level of VEGF-A121 involved in angiogenesis with poor tumor return, i.e., high VEGF-A121/165 values, may have tumors that are hypoxic and resistant to ICI treatment.

This study had several limitations. The sample size was small and, consequently, a multivariable analysis was not performed as a relatively low statistical power would weaken the confidence and reliability in determining all prognosis and treatment response-related parameters. Nonetheless, the univariable analyses provided here can be used to identify variables for inclusion in future multivariable studies; for example, the HR and RR values listed in Table 2 and Table 4 that are appreciatively different than 1.00 are strong candidates for multivariable studies. The study included only gastric cancer, not other cancer types, such as hepatocellular carcinoma, which respond well to combination therapy with VEGF inhibitors and ICIs. In addition to malignancies, a variety of diseases with increased VEGF expression have been reported, including age-related macular degeneration [43], diabetes [44], and chronic kidney disease [45]. It is possible that these co-morbidities, if present in the subject patients, might impact the serum levels of VEGF-A and VEGF-C. In this study, we did not verify the effect of co-morbidities because we did not confirm these comorbidities in detail in the subject patients.

Despite its limited sample size, this study is the first to show that the serum level of VEGF-A isoforms may be valuable as a biomarker to predict the response to ICI therapy, and assists in clarifying the causes for the failure to use VEGF-A as a definitive marker in previous studies.

## 5. Conclusions

The VEGF-A121/165 ratio may represent a new and easily measured biomarker for predicting the therapeutic response to ICIs.

## Figures and Tables

**Figure 1 cancers-16-03958-f001:**
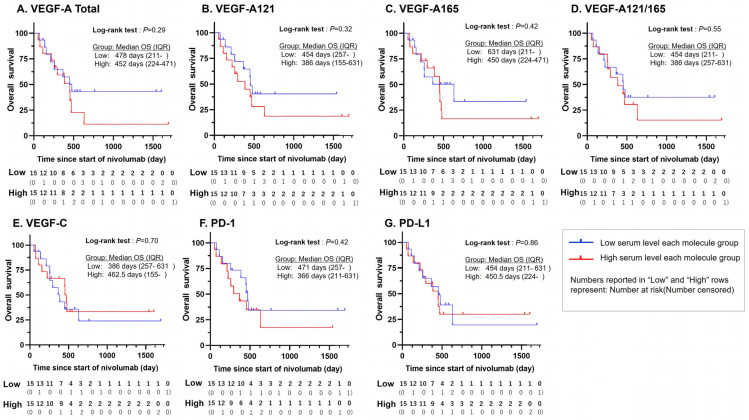
Kaplan–Meier curves of OS for each protein in patients with gastric cancer treated with nivolumab. The median OS time in the lower-level group was prolonged by (**A**) +26 days for total-VEGF-A, (**B**) +68 days for VEGF-A121, (**C**) +181 days for VEGF-A165, (**D**) +68 days for VEGF-A121/165 ratio, (**F**) +105 days for PD-1, and (**G**) +3.5 days for PD-L1 and shortened by (**E**) −76.5 days for VEGF-C, but without any case exhibiting statistical significance. Median OS times and *p*-values of the log-rank tests are given in the figure. OS—overall survival; VEGF—vascular endothelial growth factor; VEGF-A121/165, VEGF-A121-to-VEGF-A165 ratio; PD-1—programmed cell death protein 1; PD-L1—programmed death-ligand 1.

**Figure 2 cancers-16-03958-f002:**
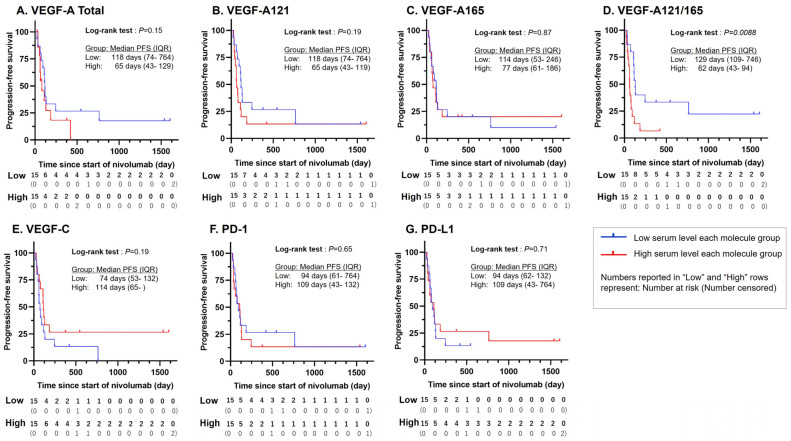
Kaplan–Meier curves of PFS for each protein in patients with gastric cancer treated with nivolumab. The median PFS time in the lower-level group was prolonged by (**A**) +53 days for total-VEGF-A, (**B**) +53 days for VEGF-A121, (**C**) +37 days for VEGF-A165, and (**D**) +67 days for VEGF-A121/165 ratio, and shortened by (**E**) −40 days for VEGF-C, (**F**) −15 days for PD-1, and (**G**) −15 days for PD-L1. Statistical significance was found only for the VEGF-A121/165 ratio, whose log-rank test *p*-value = 0.0088. All other *p*-values and all median PFS times are given in the figure. PFS—progression-free survival; VEGF—vascular endothelial growth factor; VEGF-A121/165, VEGF-A121-to-VEGF-A165 ratio; PD-1—programmed cell death protein 1; PD-L1—programmed death-ligand 1.

**Table 1 cancers-16-03958-t001:** Clinicopathological characteristics of the patients.

Clinical and Pathological Factors	*n* = 30 (%)
Sex	
Male	22 (73.3)
Age	
Median (Range)	65 (35–78)
PS	
0	11 (36.7)
1	19 (63.3)
Borrmann type	
Type 4	5 (16.7)
Dissemination	
Yes	11 (36.7)
Liver metastasis	
Yes	6 (20)
Distant lymph node metastasis	
Yes	11 (36.7)
Histological diagnosis	
Undifferentiated type	23 (76.7)
Number of nivolumab treatment courses	
Median (Range)	5.5 (1–26)
Number of chemotherapies pre nivolumab treatment	
2	23 (76.7)
≧3	7 (23.3)
Ramucirumab previous treatment	
Yes	27 (90)

**Table 2 cancers-16-03958-t002:** Risk analysis for each clinicopathological and biochemical factor in disease progression using Cox hazard analysis.

	Univariable Analysis		
	HR	95%CI	*p*-Value
Clinical and pathological factors			
Age	0.99	[0.95, 1.03]	0.65
ECOG PS, 0	1.09	[0.47, 2.44]	0.84
Histological diagnosis, Undifferentiated type	1.6	[0.64, 4.83]	0.33
Ramucirumab previous treatment, Yes	1.08	[0.32, 6.74]	0.92
Liver metastasis, Yes	0.93	[0.30, 2.35]	0.88
Distant lymph node metastasis, Yes	1.83	[0.79, 4.08]	0.16
Dissemination, Yes	1.88	[0.80, 4.38]	0.14
Serum Biomarker			
VEGF-A total	1.00	[1.00, 1.01]	0.33
VEGF-A 121 type	1.00	[1.00, 1.00]	0.27
VEGF-A 165 type	1.00	[0.99, 1.01]	0.75
VEGF-A 121/165 ratio	1.08	[1.00, 1.17]	0.0454
VEGF-C	1.00	[1.00, 1.00]	0.52
PD-1	1.00	[1.00, 1.00]	0.76
PD-L1	1.00	[1.00, 1.02]	0.17
Pre-treatment blood test values			
mGPS, 0 vs. 1–2	0.83	[0.54, 2.68]	0.65
NLR	0.96	[0.04, 5.40]	0.72
PLR	1.00	[0.99, 1.01]	0.45
LMR	1.05	[0.80, 1.36]	0.71

**Table 3 cancers-16-03958-t003:** Therapeutic response to nivolumab treatment for patients in each group.

	All Patients			Target Lesion		
	Low 121/165 Group (*n* = 15)	High 121/165 Group (*n* = 15)	*p*-Value	Low 121/165 Group (*n* = 8)	High 121/165 Group (*n* = 11)	*p*-Value
Best overall response, *n* (%)						
Complete response	3 (20)	0 (0)		2 (25)	0 (0)	
Partial response	2 (13)	0 (0)		2 (25)	0 (0)	
Stable disease	3 (20)	2 (13)		3 (38)	2 (18)	
Progressive disease	3 (20)	10 (67)		1 (12)	9 (82)	
non-CR/non-PD	4 (27)	3 (20)		-	-	
Objective Response Rate (%)	-	-		50	0	0.0181
Disease Control Rate (%)	80	33	0.0253	88	18	0.0055

**Table 4 cancers-16-03958-t004:** Association of each clinicopathological and biochemical factor with disease progression based on logistic single regression analysis.

	Univariable Analysis		
	RR	95%CI	*p*-Value
Clinical and pathological factors			
Age	1	[0.93, 1.08]	0.95
ECOG PS, 0	1.15	[0.25, 5.20]	0.86
Histological diagnosis, Undifferentiated type	6.55	[0.68, 63.33]	0.06
Ramucirumab previous treatment, Yes	0.34	[0.01, 4.02]	0.39
Liver metastasis, Yes	1.4	[0.22, 9.01]	0.71
Distant lymph node metastasis, Yes	3.79	[0.83, 19.81]	0.09
Dissemination, Yes	1.15	[0.25, 5.20]	0.86
Serum Biomarker			
VEGF-A total	1.00	[1.00, 1.01]	0.0032
VEGF-A 121 type	1.00	[1.00, 1.01]	0.0043
VEGF-A 165 type	1.00	[0.98, 1.02]	0.96
VEGF-A 121/165 ratio	1.2	[1.01, 1.38]	0.0319
VEGF-C	1.00	[0.99, 1.00]	0.63
PD-1	1.00	[0.99, 1.00]	0.73
PD-L1	1.02	[1.00, 1.06]	0.06
Pre-treatment blood test values			
mGPS, 0 vs. 1–2	0.96	[0.24, 4.51]	0.96
NLR	1.06	[0.71, 1.61]	0.77
PLR	1.00	[0.004, 3.60]	0.26
LMR	0.79	[0.44, 1.31]	0.37

## Data Availability

The data presented in this study are available on request from the corresponding author.

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
