# Peer review of "The Vascular Endothelial Growth Factor-A121/Vascular Endothelial Growth Factor-A165 Ratio as a Predictor of the Therapeutic Response to Immune Checkpoint Inhibitors in Gastric Cancer"

_cancers, 2024, doi:10.3390/cancers16233958_

Round 1
Reviewer 1 Report
Comments and Suggestions for Authors
The article „ VEGF-A121-to-VEGF-A165 ratio as a predictor of the therapeu-2 tic response to immune checkpoint inhibitors in gastric cancer” highlights the role of biomarkers for predicting the response to immune checkpoint inhibitors (ICIs) the treatment of many cancers This was an interesting article, and I read it with interest. I think it is important in cancer’s biomarkers and future treatments field.
I suggest a few minor revisions. Punctuation and spelling are required, and English speaker should revise the manuscript. References should respect journal requirements.
Comments on the Quality of English LanguageThe article „ VEGF-A121-to-VEGF-A165 ratio as a predictor of the therapeu-2 tic response to immune checkpoint inhibitors in gastric cancer” highlights the role of biomarkers for predicting the response to immune checkpoint inhibitors (ICIs) the treatment of many cancers This was an interesting article, and I read it with interest. I think it is important in cancer’s biomarkers and future treatments field.
I suggest a few minor revisions. Punctuation and spelling are required, and English speaker should revise the manuscript. References should respect journal requirements.
Author Response
Comments 1: Punctuation and spelling are required, and English speaker should revise the manuscript. References should respect journal requirements.
Response 1: Co-author Drew N. Maywar, an English speaker, reviewed the manuscript again and picked out the English grammar errors. The corrections made according to his suggestions are indicated by green highlighting {Page 1, Line 16, 18, 21, 39, Page 2, Line 48, 49, 50, 59, Page 3, Line 104, 129, 134, 135, 136, 137, 141, 142, 143, Page 5, Line 183, 189, Page 6, Line 200 (Table 2), Page 7, Line 213, 214, 215, Line 216 (Table 3), Line 218 (Table 4), Page 8, Line 231, 244, 247, 260, Page 9, Line 288, 289, 290, 293}. We further confirmed again that the references complied with the journal's requirements.
Reviewer 2 Report
Comments and Suggestions for Authors
In general, the manuscript “VEGF-A121-to-VEGF-A165 ratio as a predictor of the therapeu-2 tic response to immune checkpoint inhibitors in gastric cancer” is well written and overall provides a good summary and explanation of the research problem and the results.
The justification of the study is really well pointed out. It explains the lack of information and gives the solution to the problem (assay develop). However, I wonder why only gastric cancer samples were analyzed and do not include samples from hepatocellular carcinoma where VEGF inhibitors show good results against it as authors mentioned
The methodology and results are presented in a clear and simple manner, although I think the data is inconclusive in pointing out a clear correlation of VEGF serum levels and cancer prognosis. As you correctly stated in your conclusion section.
I think you also need to include controls on your experiment design. Beig serum of healthy individuals to set the baseline of normal values.
I will only suggest increasing your sample size adding a variety of cancers samples. This will make the study and data generated more robust.
My Overall recommendation is to Accept after minor revisions
Author Response
Comments 1: I wonder why only gastric cancer samples were analyzed and do not include samples from hepatocellular carcinoma where VEGF inhibitors show good results against it as authors mentioned.
Response 1: Hepatocellular carcinoma was not selected for this study because the use of ICI for hepatocellular carcinoma was first approved in Japan in September 2020, the number of patients was not yet available, and data on prognosis and therapeutic efficacy would be insufficient. However, we agreed with reviewer #2 and therefore added in the limitation of the Discussion that the study was performed only in gastric cancer (red highlighted, Page 9, Line 280-282).
Comments 2: I think you also need to include controls on your experiment design. Beig serum of healthy individuals to set the baseline of normal values.
Response 2: Thank you for pointing this out. We had measured VEGF-A121 and VEGF-A165 serum levels in healthy controls in a previous study (reference 20) and set the values found in that study as a baseline. We have added a description of the controls in the Methods section of our manuscript (red highlighted, Page 3, Line 107-109).
Comments 3 I will only suggest increasing your sample size adding a variety of cancers samples. This will make the study and data generated more robust.
Response 3: Agree. Although we did not perform multivariable analysis due to the small sample size, we noted that some variables in our univariable analysis could be candidates for multivariate analysis in the future when we increase the sample size (red highlighted, Page 9, Line 277-280).
Reviewer 3 Report
Comments and Suggestions for Authors
Rapid selection of an adequate cancer treatment strategy is one of the most important factors for successful patient treatment, as it is known that some drugs that have shown very good results in some patients do not give the same result in others, this happens due to various factors that must be taken into account to optimize treatment and save a very important resource - time! Within the framework of this article, the authors collected information and analyzed clinical data, as a result, it was found that the serum ratio of vascular endothelial growth factor VEGF-A121/165 may be a new biomarker for predicting therapeutic response to ICI, which can be easily measured. I believe that this article is of significant interest to practicing oncologists and fully meets the criteria of the Cancers journal.
Author Response
We could not find anything in reviewer #3's comments that would indicate a modification on our manuscript.
Reviewer 4 Report
Comments and Suggestions for Authors
The manuscript titled “VEGF-A121-to-VEGF-A165 Ratio as a Predictor of the Therapeutic Response to Immune Checkpoint Inhibitors in Gastric Cancer” presents a very good idea and is well-written. However, I have a few minor questions regarding the methods section, which are as follows:
The writing of the statistical analysis session seems to be from a non-statistician. I would like to suggest two comments for the manuscripts re. data analysis.
1. Please have an experienced statistician review your statistical methods. Some statistical terms appear to be used incorrectly, which could lead to confusion. For example, the phrase “…using univariate analysis with a Cox proportional hazards model” and “Univariate analysis of the correlation with treatment response was performed via logistic single regression analysis” need clarification.
- Please clarify how the assumptions of the Cox proportional hazards model were evaluated.
- What are the normal controls (without any disease condition) for this study?
- Is there any correlation between age and upregulation of VEGF A and VEGF C and other co-morbidity? Please cite some previous research
Author Response
Comments 1: Please have an experienced statistician review your statistical methods. Some statistical terms appear to be used incorrectly, which could lead to confusion. For example, the phrase “…using univariate analysis with a Cox proportional hazards model” and “Univariate analysis of the correlation with treatment response was performed via logistic single regression analysis” need clarification.
Response 1: We asked co-author Drew N. Maywar (a expert in statistics) to review again the statistical descriptions in our manuscript. He corrected several terminological errors. They are highlighted in yellow. {Page 1 Line 19, 29, 33, 34, Page 2 Line 85, Page 3 Line 117, 121, 122, 123, 126, 128, Page 4 Line 152, Page 5 Line 186, Page 6 Line 200 (Table 2), Page 7 Line 203, 211, 212, 218 (Table 4), Page 8, Line 247, Page 9, Line 275, 288}.
Comments 2: Please clarify how the assumptions of the Cox proportional hazards model were evaluated.
Response 2: The univariable analyses provided in this study can be used to identify variables for inclusion in future multivariable studies; for example, the HR and RR values listed in Tables 2 and 4 that are appreciatively different than 1 are strong candidates for multivariable studies. We added this evaluation to the Discussion (red highlighted, Page 9 Line 277-280).
Comments 3: What are the normal controls (without any disease condition) for this study?
Response 3: We had measured VEGF-A121 and VEGF-A165 serum levels in healthy controls in a previous study (reference 20) and set the values found in that study as a baseline. We have added a description of the controls in the Methods section of our manuscript (red highlighted, Page 3, Line 107-109).
Comments 4: Is there any correlation between age and upregulation of VEGF A and VEGF C and other co-morbidity? Please cite some previous research
Response 4: In besides malignancies, a variety of diseases with increased VEGF expression have been reported, including age-related macular degeneration, diabetes, and chronic kidney disease. It is possible that these co-morbidities, if present in the subject patients, might impact the serum levels of VEGF-A and VEGF-C. In this study, we did not verify the effect of co-morbidities because we did not confirm these co-morbidities in detail in the subject patients. We added a description of this limitation (red highlighted, Page 9, Line 274, 282-287, Page 11, Line 431-436). We searched for studies in cancer models that explored the potential of VEGF-A or VEGF-C as biomarkers, as in this study, but could not find any reports showing a significant correlation between age and VEGF expression.
Round 2
Reviewer 4 Report
Comments and Suggestions for Authors
The author's thoroughly addressed all comments, and manuscript is ready for publication.